# Acaricidal activity of plant extracts against *Amblyomma variegatum* in Waghimra, Northeastern Ethiopia

**Teklu Yitbarek[1]**, **Abebe Tibebu** **[1]\***, **Adane Bahiru[1]**, **Ayalew Assefa[2]**

**1** Amhara Agricultural Research Institute, Sekota Dryland Agricultural Research Center, Sekota, Ethiopia,
**2** International Livestock Research Institute (ILRI), Addis Ababa, Ethiopia

☯ Teklu Yitbarek and Abebe Tibebu contributed equally to this work.
\* abetfek@gmail.com

## Abstract

Ticks are blood-feeding ectoparasites that cause significant economic losses by reducing animal productivity and damaging hides and skins. Ticks control and prevention strategies primarily rely on the regular use of synthetic acaricides. However, the widespread misuse of these commercial acaricides has led to the emergence of acaricide-resistant ticks, compromising their sustainable effectiveness. Therefore, this study aimed to evaluate alternative plant-based acaricides and determine their effective dose levels. Crude ethanol extracts from six plant species, previously reported to possess acaricidal activity, were initially screened at a concentration of 100 mg/mL using the adult immersion test. Based on the prescreening results, *Ricinus communis*, *Azadirachta indica*, and *Jatropha curcas*, which showed promising acaricidal effects, were subsequently tested at concentrations of 12.5, 25, 50, and 100 mg/mL. Tick mortality and oviposition inhibition were monitored over seven days at 3-hour intervals. *Ricinus communis* produced the highest acaricide efficacy ($70 \pm 1.8\%$) at 100 mg/mL. In comparison, 100 mg/mL of *Azadirachta indica* and *Jatropha curcas* resulted in $56.7 \pm 2.2\%$ and $16.7 \pm 3.02\%$ acaricide activity, respectively. The minimum lethal doses required to kill 50% ($LD_{50}$) and 90% ($LD_{90}$) of the tested ticks were 18.47 mg/ml and 184.3 mg/ml for *Ricinus communis* and 83.4 mg/ml and 431.8 mg/ml for *Azadirachta indica*, respectively. *Ricinus communis* exhibited the highest mean oviposition inhibition (98.5%), followed closely by *Azadirachta indica* (96.57%), while *Jatropha curcas* demonstrated comparatively lower inhibition (68.05%). A statistically significant dose-dependent oviposition inhibition effect was observed for *Ricinus communis* and *Azadirachta indica* ($p < 0.05$). In conclusion, *Ricinus communis* and *Azadirachta indica* demonstrated strong potential as botanical acaricide alternatives and warrant further evaluation in clinical trials.

**Data availability statement:** Data used for this study is submitted with this paper.

**Funding:** We thank the Amhara Region Agricultural Research Institute (ARARI) for the financial and logistical support. The funders had no role in the study design, data collection and analysis, decision to publish, or preparation of the manuscript.

**Competing interests:** The authors have declared that no competing interests exist.

## Introduction

Ticks are obligate blood-feeding ectoparasites that cause blood loss, irritation, and toxicosis. Tick infestations' associated economic losses include reduced weight gain, lower milk production, poor hide quality, and increased veterinary costs [1]. Ticks can transmit pathogens, including viruses, bacteria, and protozoa, leading to anaplasmosis, babesiosis, theileriosis, and heartwater diseases between animals [2]. In endemic areas, ticks maintain and transmit East Coast fever between animals, a transboundary disease with significant impact in livestock production [3].

*Amblyomma variegatum* (*A. variegatum*) is one of the most economically important and widely distributed livestock tick species in sub-Saharan Africa. In Ethiopia, it is the most common tick species, particularly in the Wag-Lasta area [3–5]. It causes severe bite wounds often complicated by secondary infections and serves as a major vector of pathogens such as *Ehrlichia ruminantium* and *Rickettsia africae*, the causative agents of heartwater and African tick-bite fever, respectively [6].

For a long time, ticks have been controlled using chemical acaricides; however, numerous reports indicate increasing resistance to commercial formulations due to improper application, overdosing, and lack of acaricide rotation. Various classes of acaricides, including organophosphates and pyrethroids, are becoming less effective due to the development of resistance by many of economically important tick species [7,8]. *Amblyomma variegatum* has shown signs of resistance to diazinon and amitraz in the Waghimra area [4], where rural farmers face constraints such as limited access, high costs, and poor availability of alternative commercial acaricides.

The increasing development of resistance in tick populations to conventional chemical acaricides has reduced the effectiveness of existing control measures, thereby driving the use of plant-based acaricides, which are gaining increasing popularity. In Ethiopia, several indigenous plant species, including *Azadirachta indica* (*A. indica)*, *Phytolacca dodecandra*, *Calpurnia aurea*, and *Millettia ferruginea*, are traditionally used for ectoparasites control and possess potential acaricidal properties [9–11]. Similarly, in North Shewa, Ethiopia, traditional medicinal plants, including *Allium sativum*, *Ricinus communis* (*R. communis*), and *Calotropis procera* (*C. procera*), have been widely employed for ectoparasites control [12]. The widespread availability, cultural acceptance, and potential for cultivation of these plant species make them promising candidates for the development of community-based, sustainable tick management strategies.

To the best of the authors' knowledge, although several indigenous plants have been traditionally used in the study area [10] and elsewhere to control tick infestations, their acaricidal efficacy has not yet been experimentally investigated. To address this gap, this study was initiated to evaluate plant species with potential acaricidal properties and to determine their effective dosage levels using the Adult Immersion Test (AIT) protocol. The AIT has been widely used to evaluate the efficacy of acaricides and remains a reliable and valid method for testing both natural and synthetic agents [13].

 

## Materials and methods

### Study area

The study was conducted in the Sekota district of Waghimra Zone, Amhara, Ethiopia. The district is characterized by a midland agroclimatic zone, with elevations ranging from 2,000–2,300 meters and an average annual rainfall of 774.3 mm. The area experiences a main rainy season from late June or early July, lasting a few weeks to a month, though in some years it may extend longer. Mixed farming is the primary agricultural system, with cattle, goats, and sheep raised under an extensive production system. Major crops cultivated include wheat, barley, sorghum, and various pulses. The villages included in the tick collection survey were Sereul, Tsemera, Rubaria, and Keva.

### Study animal and tick collection

The target population for tick sample collection comprised cattle that had not been treated with commercial acaricides for at least one month. A cross-sectional field tick collection survey was conducted from May to June 2023 in the selected villages. All fully engorged adult female ticks were carefully detached from naturally infested cattle using forceps and transported to the Sekota Dryland Agricultural Research Center Veterinary Laboratory (SDARC-VL) in clean, dry, well-labeled plastic containers, ensuring sufficient ventilation during transport. Upon arrival at the SDARC-VL laboratory, ticks were identified to genus and species levels using established taxonomic keys and a stereomicroscope. Identification focused on *A. variegatum*, one of the most abundant and economically important tick species in the study area, which was selected as the experimental species [4,5]. Accordingly, approximately 600 *A. variegatum* ticks were retained and used for the experimental study.

### Plant collection, processing and crude extraction

Based on a previous ethnoveterinary medicinal plant survey and evidence from other studies, six plant species, Commiphora africana (root), *R. communis* (seed), *C. procera* (juice), *A. indica* (leaf), Allium sativum (bulb), and Jatropha curcas (leaf), were selected and collected [10]. Young leaves of *A. indica* and *Jatropha curcas* were harvested from Rubaria (Sekota District), while the root of *Commiphora africana* was collected from Bilaku (Ziquala District). Fresh bulbs of *Allium sativum* were purchased from the local market, and seeds of *R. communis* were collected from Woleh (Sekota District). Additionally, fresh leaves of *C. procera* were collected from Tsemera and Keva villages (Sekota District) (Table 1).

Preprocessing of the collected plant materials was carried out at the SDARC-VL, following standard procedures. Fresh leaves of *A. indica* and *Jatropha curcas* were washed with running water, dried under shade, and stored at room temperature. The bulbs of *Allium sativum*, the root of *Commiphora africana*, and the seeds of *R. communis* were washed, chopped and dried, whereas the juice of *C. procera* was allowed to dry and crushed. All plant materials were allowed to dry and then ground using a blender and sieved to obtain fine powders.

Finally, 500 grams of each powdered plant material were packed and submitted to the Wollo University Chemistry Laboratory for subsequent active ingredient extraction procedures using the maceration technique. Briefly, each powdered

**Table 1. Crude ethanolic extract yield (gram) of selected plant materials.**

| Scientific Name | Family | Plants part | Source | Dry weight(g) | Yield (g) | Yield (%) |
|---|---|---|---|---|---|---|
| *Ricinus communis* | Euphorbiaceae | Seed | Sekota | 500 | 45.6 | 9.1 |
| *Allium sativum* | Alliaceae | Bulb | Sekota | 500 | 2.5 | 0.5 |
| *Azadirachta indica* | Meliaceae | Leaf | Sekota | 500 | 15.4 | 3.1 |
| *Jatropha curcas* | Euphorbiaceae | Leaf | Sekota | 500 | 20.5 | 4.1 |
| *C. africana* | Burseraceae | Root | Ziquala | 500 | 17.4 | 3.5 |

plant material was macerated in 1300 mL of 95% ethanol in a tightly sealed flask and kept at room temperature for 72 hours to facilitate the dissolution of bioactive compounds, with frequent agitation, as described by Arshed et al. [14]. The crude extracts were subsequently filtered using Whatman filter paper, transferred into light-proof glass containers, and stored at +4 °C until the commencement of the experiment. The percentage extraction yield of each plant material was estimated as follows: calculated [15]:

$$\text{Extraction rate (\%)} = ((\text{Final extract weight}) / (\text{Initial extract weight})) \times 100.$$

## Application methods and bioassay evaluation

The tickicidal experiment was conducted using a completely randomized design (CRD), employing the adult tick immersion test (AIT) in accordance with the standard procedures described in the FAO [13] guidelines. Initially, seven groups, each comprising ten fully engorged adult female ticks (70 ticks in total), were randomly selected, weighed, and immersed in 10 ml of each plant extract at the highest concentration (100 mg/ml), as well as in distilled water (control), for 10 minutes. After immersion, the ticks were sieved, gently dried using paper towels, and finally placed in labeled Petri dishes and followed for 7 days. Three plant extracts that demonstrated significant mortality and egg-laying inhibition during this screening stage were selected for further dose-dependent evaluation.

In the subsequent dose-dependent evaluation, each treatment was tested at four different concentrations with three replicates each (Table 2). Distilled water was used as a negative control. A total of 390 treated ticks, allocated into 39 experimental units of 10 ticks each, were mounted dorsally in Petri dishes using double-sided adhesive tape. Treated ticks were then incubated at 27 ± 1°C with relative humidity maintained above 80%, and observed at 3-hour intervals for seven consecutive days. At each observation, tick mortality, reflex responses to mechanical stimulation, and body coloration were recorded. Ticks that exhibited movement and responded to needle touch stimuli were considered alive, whereas those showing no movement, inability to maintain posture or leg coordination, or failure to reposition themselves were considered dead.

After seven days of incubation, final live ticks were counted, and the mass of eggs laid in each treatment and concentration group was weighed. Antiparasitic efficacy (AE), representing the acaricidal effect of the treatments on tick survival, was calculated as:

$AE = B-T/ B \times 100$, where B is the mean survival of ticks in the control group, T is the mean survival of ticks in the treatment group, and AE represents the antiparasitic efficacy [4]. The Egg-Laying Test (ELT) approach was used to evaluate the effect of treatments on oviposition inhibition. The percentage inhibition of oviposition (percent control) was calculated as:

$$PC = (MEC - MET) / (MET) \times 100,$$

where MEC is the mean egg mass of control ticks and MET is the mean egg mass of treated ticks [13].

**Table 2. Plant extracts tested for dose-dependent evaluation of acaricidal activity (12.5, 25, 50 and 100 g/ml).**

| Treatments | Extracts/Control |
|---|---|
| Treatment 1 | *Negative Control* |
| Treatment 2 | *Ricinus communis* |
| Treatment 3 | *Jatropha curcas* |
| Treatment 4 | *Azadirachta indica* |

## Ethics approval

All animal sampling procedures complied with the ARRIVE guidelines and were conducted in accordance with the U.K. Animals (Scientific Procedures) Act, 1986. The sampling was performed as part of routine veterinary management practices, and since no endangered or protected species were involved, specific ethical clearance or official permission was not required. Furthermore, the researcher clearly explained the objectives of the study to the farmers prior to their participation in field tick collection. Participants were informed that their involvement was voluntary and confidential and that they could withdraw at any time during the tick and data collection.

## Statistical analysis

Data were entered, cleaned, coded, and edited using Microsoft Excel. All statistical analyses were performed in STATA version 17 (StataCorp, Texas, USA). Tick mortality was summarized descriptively in percentages and means with standard deviations. The tickicidal effects on oviposition inhibition by each plant extract at different concentrations were analyzed using analysis of variance (ANOVA). The mean lethal doses ($LD_{50}$) and 90% lethal doses ($LD_{90}$) were estimated using probit analysis. A p-value $< 0.05$ was considered statistically significant.

## Result and discussion

### Acaricidal efficacy (AE) of plant extracts

The mean survival and acaricidal efficacy of the three tested plant extracts was presented in Table 3. Ticks treated with *R. communis* demonstrated the lowest mean survival and the highest acaricidal efficacy, whereas those treated with *Jatropha curcas* exhibited the highest mean survival and the lowest acaricidal efficacy. The present findings were consistent with previous studies demonstrating the effectiveness of *R. communis* and *A. indica* against ticks [16]. *Jatropha curcas* has also shown acaricidal activity against spider mites, although its efficacy was lower than that of *R. communis* according to the report by Ghorab & Ismail [17], which aligns with the results of the current study.

The dose-dependent acaricidal efficacy showed that *R. communis* exhibited the highest tick mortality at its maximum concentration. Similarly, *A. indica* and *Jatropha curcas* also exhibited the highest tick mortality at their respective maximum concentrations. *Ricinus communis* showed a progressive increase in tick mortality with increasing concentration up to 50 mg/mL, followed by a slight decline towards the final highest concentration. *A. indica* showed a consistent increasing tick mortality across its concentration gradients. In contrast, *Jatropha curcas* exhibited an inconsistent pattern, with the highest mean tick survival observed at 50 mg/ml and its maximum tick mortality achieved at 100 mg/ml (Fig 1).

The present study also revealed substantial variation in plant extract acaricidal efficacy across different concentrations. *R. communis* achieved 70±1.83% acaricidal efficacy (AE), followed by *A. indica* with an AE of 56.7±2.2% at the respective highest dose (Table 4). This was consistent with a study conducted in Ethiopia, where the ethanolic extract of *R. communis* showed 73% AE against *Rhipicephalus pulchellus* [15]. The findings were also consistent with other reports elsewhere that the highest tick mortality with *R. communis* was achieved at its maximum concentration [18,19]. A study in

**Table 3. Mean tick mortality and acaricidal efficacy (AE) of plant extracts.**

| Treatments | Mean death | AE (%)±SD |
|---|---|---|
| Distilled water | 0.0 | |
| *Ricinus communis* | 5.83 | 58.3±0.01% |
| *Jatropha curcas* | 1.0 | 10.0±13.22% |
| *Azadirachta indica* | 4.5 | 45.0±4.29% |

SD = Standard deviation, AE = Acaricidal efficacy.

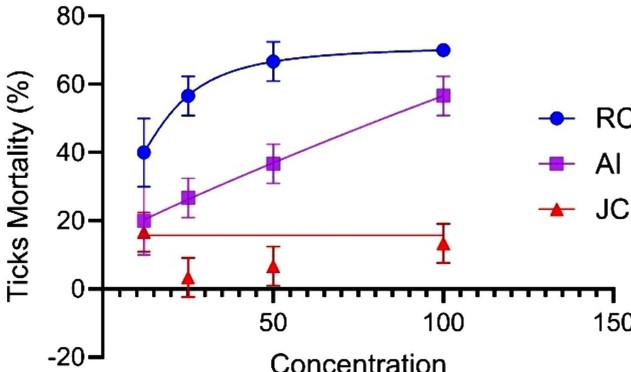

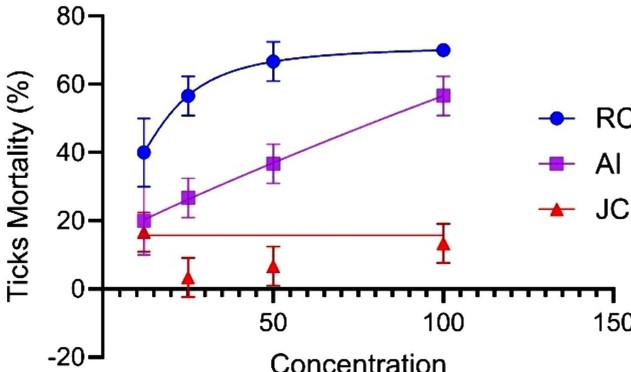

**Fig 1. Dose-dependent mortality of ticks treated by of plant extracts (RC = *Ricinus communis*, AI = *Azadirachta indica*, JC = *Jatropha curcas*).**

**Table 4. Dose dependent plant extracts Acaricidal efficacy (AE).**

| Treatments | Dose(mg/ml) | Mean Death | AE (%)±SD |
|---|---|---|---|
| *Ricinus communis* | 100 | 7 | 70 ± 1.83% |
| | 50 | 6.66 | 66.7 ± 1.93% |
| | 25 | 5.66 | 56.7 ± 2.20% |
| | 12.5 | 4.0 | 40 ± 2.58% |
| *Jatropha curcas* | 100 | 1.77 | 16.7 ± 3.02% |
| | 50 | 0.33 | 3.3 ± 3.28% |
| | 25 | 0.66 | 6.7 ± 3.22% |
| | 12.5 | 1.33 | 13.3 ± 3.10% |
| *Azadirachta indica* | 100 | 5.66 | 56.7 ± 2.20% |
| | 50 | 4.66 | 36.7 ± 2.43% |
| | 25 | 2.66 | 26.7 ± 2.90% |
| | 12.5 | 2.0 | 20 ± 72.98% |

SD = Standard deviation, AE = Acaricidal efficacy.

Bangladesh reported that ethanolic extracts of *R. communis* maintained notable acaricidal potential even at lower concentrations (0.45 mg/15 ml), although its efficacy decreased compared to higher doses, aligning with the findings of Islam et al. [20]. Moreover, the high dose (100 mg/ml) of *R. communis* demonstrated a comparable AE with commercial acaricide of Amitraz (57.5%), as reported by Tibebu and Assefa [4].

The acaricidal efficacy observed in the present study was also comparable with findings from other plant extracts evaluated against different external parasites. A study conducted in Egypt showed that an extract of *Alcea rosea* at 150 mg/ml produced an 85% direct killing effect against *Tetranychus urticae* [17], which is relatively comparable with the results of the current study.

*Azadirachta indica* showed a dose-dependent acaricidal effect on tick, consistent with earlier research indicating that higher concentrations produce stronger acaricidal efficacy. This finding aligns with reports by Blando [21] and Abu Hawsah et al. [22], which demonstrated that increasing *A. indica* concentrations enhanced tick mortality. Similarly, Shyma et al. [23] reported higher acaricidal efficacy of *A. indica* in India, and Bakewell-Stone [24] highlighted that its efficacy varies significantly depending on the application method.

On the contrary, *Jatropha curcas* exhibited the lowest AE in comparison to both *A. indica* and *R. communis*. This was consistent with the finding of that lower efficacy against the *Tetranychus urticae* compared to *R. communis* [17].

In the present study a significant increase in tick mortality started at 3 hr post exposure with concentrations of 100 and 50 mg/ml *R. communis* extract, while the study by Kemal [15] reported 30 min for 100 mg/ml; 2 hr for 50 and 25 mg/ml; and 3 hr for 12.5 mg/ml post exposure of *R. pulchellus*. The highest toxicity against ticks was due to the presence of bioactives in the extract of *R. communis* [25].

## Plant extracts egg laying inhibition efficacy

Table 5 illustrates the *R. communis* and *A. indica* treatments resulted in a significant inhibition of oviposition, 98.65% and 96.57%, respectively. The high percent control (PC) value suggests a strong impact on tick reproductive success, making it a promising candidate for further exploration as an acaricidal agent. Both *R. communis* and *A. indica* exhibited greater acaricidal efficacy compared to *Jatropha curcas* (68.05%). The inhibition of oviposition at the highest concentration of *A. indica* was reported to be 20.73% [23], which is lower than the value observed in the present study.

Table 6 presents an analysis of variance evaluating the egg-laying inhibition effects of the different plant extracts, with emphasis on treatment, dose levels, and their interaction. The result indicated that there was statistically highly significant difference in egg-laying inhibition effect ($p < 0.001$). Among the sources of variation, treatment had a strong and statistically significant effect ($p < 0.001$), indicating that the plant extracts differ markedly in their inhibitory potential. In contrast, the effects of dose and the interaction between treatment and dose were not significant. The low residual mean square (0.056) further supports the reliability of the model's findings.

## Minimum lethal dose to cause 50% and 90% mortality

Probit regression analysis was performed to estimate the median lethal concentration ($LD_{50}$) and ninety percent lethal concentration ($LD_{90}$) of three plant extracts against adult *A. variegatum* ticks. A statistically significant positive dose

Table 5. The mean comparison of oviposition (g) and percent control (PC).

| Treatments | Mean Egg Mass | PC (%) |
|---|---|---|
| *Ricinus communis* | 0.03ac | 98.65 ± 0.7% |
| *Jatropha curcas* | 0.69bc | 68.05 ± 4.5% |
| *Azadirachta indica* | 0.07c | 96.57 ± 1.8% |
| Distilled water | 2.17d | |

PC = Percent control, SD = Standard deviation.

Table 6. ANOVA of egg-laying inhibition effect of plant extracts.

| Source | SS | Df | MS | F | Prob>F |
|---|---|---|---|---|---|
| Model | 13.61 | 12 | 1.13 | 20.40 | 0.000 |
| Treatment | 11.95 | 3 | 3.98 | 71.67 | 0.000 |
| Dose | 0.083 | 3 | 0.028 | 0.50 | 0.686 |
| Treatment # Dose | 0.22 | 6 | 0.037 | 0.66 | 0.681 |
| Residual | 1.45 | 26 | 0.056 | | |
| Total | 15.05 | 38 | 0.40 | | |

DF = Degrees of freedom, F = F-ratio, MS = Mean square, SS = Sum of squares.

dependent mortality relationship was observed for *R. communis* ($\beta = 0.008$, p = 0.029). The estimated $LD_{50}$ and $LD_{90}$ concentrations were 18.48 mg/mL and 184.33 mg/mL, respectively, indicating that relatively low concentrations of the extract were required to achieve 50% mortality, while substantially higher concentrations were needed to attain 90% mortality. The lower 95% confidence interval for $LD_{50}$ (35.34 mg/mL to 48.23 mg/mL) indicates minimum variability, whereas the larger LD90 confidence interval (114.93 mg/mL to 1425.57 mg/mL) suggests variability in response at higher concentrations.

Similarly, *A. indica* demonstrated a highly significant dose–response relationship ($\beta = 0.011$, SE = 0.004, p = 0.002). The $LD_{50}$ and $LD_{90}$ concentrations were estimated at 83.43 mg/mL and 198.65 mg/mL, respectively, which was consistent with the finding by Kumar et al. [26]. The 95% confidence interval for $LD_{50}$ ranged from 60.95 to 149.80, while that for $LD_{90}$ ranged from 138.78 mg/mL to 451.41 mg/mL. The relatively narrower confidence intervals indicate a stable and reliable probit model fit. In contrast, *Jatropha curcas* did not show a statistically significant dose–mortality relationship ($\beta = 0.003$, SE = 0.004, z = 0.759, p = 0.448). The estimated $LD_{50}$ and $LD_{90}$ values were 431.80 mg/mL and 814.41 mg/mL, respectively. However, confidence intervals for these estimates were not generated due to model instability, indicating that the extract exhibited weak acaricidal activity under the tested conditions (Table 7).

Overall, among the tested plant extracts, *R. communis* and *A. indica* exhibited the lowest $LD_{50}$ concentration values, indicating comparatively higher tickicidal activity, whereas *Jatropha curcas* demonstrated the lowest acaricidal activity.

## Limitation of the study

The experiment was conducted without a positive control, which limits the robustness of the estimated acaricidal efficacy for each treatment. Additionally, the reproductive index and hatchability of laid eggs were not assessed, as the experiment was originally designed to evaluate only mortality and oviposition inhibition. Therefore, caution should be given when interpreting these findings.

## Conclusion and recommendation

This study demonstrated that locally available plant extracts exhibit notable acaricidal and oviposition-inhibiting activity against *A. variegatum*. Both *R. communis* and *A. indica* showed clear dose-dependent acaricidal effects, with the highest efficacy observed at the 100 mg/mL concentration for both tick mortality and oviposition inhibition. In contrast, *Jatropha curcas* did not exhibit a dose-dependent response. Based on these findings, *R. communis* and *A. indica* are recommended for further clinical and toxicological evaluation.

**Table 7. Probit regression analysis of $LD_{50}$ and $LD_{90}$ for the three plant extracts.**

| Treatment | | Estimate(Se) | Z | Sig | LD50 | LD90 |
|---|---|---|---|---|---|---|
| RC | Concentration (%) | 0.008 (0.004) | 2.18 | 0.029 | 18.479 (35.34-48.23) | 184.333(114.92-1425) |
| | Intercept | −0.143 (0.20) | −0.721 | 0.471 | – | – |
| AI | Concentration (%) | 0.011(0.004) | 3.128 | 0.002 | 83.426(60.95-149.80) | 198.651(138.77-451.41) |
| | Intercept | −0.928(0.21) | −4.348 | 0.001 | – | – |
| JC | Concentration (%) | 0.003(0.004) | 0.759 | 0.448 | 431.797 | 814.405 |
| | Intercept | −1.446(0.27) | −5.308 | <.001 | – | – |

LD50 = lethal dose 50, LD90 = lethal dose 90.

## Supporting information

**S1 Table. Data used for analysis.**
(XLSX)

## Acknowledgments

The authors would like to express their sincere gratitude to the animal owners for their cooperation. We also wish to acknowledge the community animal health workers in the study area for their invaluable assistance in facilitating the fieldwork.

## Author contributions

**Conceptualization:** abebe tibebu, Teklu Yitbarek, Adane Bahiru.

**Data curation:** abebe tibebu, Teklu Yitbarek, Adane Bahiru.

**Formal analysis:** abebe tibebu.

**Investigation:** abebe tibebu, Teklu Yitbarek.

**Methodology:** abebe tibebu, Ayalew Assefa.

**Supervision:** Teklu Yitbarek.

**Writing – original draft:** abebe tibebu, Teklu Yitbarek, Adane Bahiru.

**Writing – review & editing:** abebe tibebu, Teklu Yitbarek, Adane Bahiru, Ayalew Assefa.

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
