## [Decision Letter · Decision Letter 0]

17 Feb 2026

PONE-D-25-64821Acaricidal Activity of Plant Extracts Against Amblyomma variegatum Tick Species in Waghimra, Northeastern EthiopiaPLOS One

Dear Dr. tibebu,

Thank you for submitting your manuscript to PLOS ONE. After careful consideration, we feel that it has merit but does not fully meet PLOS ONE’s publication criteria as it currently stands. Therefore, we invite you to submit a revised version of the manuscript that addresses the points raised during the review process.

We look forward to receiving your revised manuscript.

Kind regards,

Kandasamy Ulaganathan

Academic Editor

PLOS One

Journal Requirements:

3. Please provide additional details regarding consent obtained from the owners of the animals and mention the detailed procedure.

4. Please note that funding information should not appear in any section or other areas of your manuscript. We will only publish funding information present in the Funding Statement section of the online submission form. Please remove any funding-related text from the manuscript.

“The authors would like to express their sincere gratitude to the animal owners for their cooperation. We also wish to acknowledge the community animal health workers in the study area for their invaluable assistance in facilitating the fieldwork. Finally, we extend our thanks to the Amhara Region Agricultural Research Institute (ARARI) for the financial and logistical support.”

6. In the online submission form you indicate that your data is not available for proprietary reasons and have provided a contact point for accessing this data. Please note that your current contact point is a co-author on this manuscript. According to our Data Policy, the contact point must not be an author on the manuscript and must be an institutional contact, ideally not an individual. Please revise your data statement to a non-author institutional point of contact, such as a data access or ethics committee, and send this to us via return email. Please also include contact information for the third party organization, and please include the full citation of where the data can be found.

7. Your ethics statement should only appear in the Methods section of your manuscript. If your ethics statement is written in any section besides the Methods, please move it to the Methods section and delete it from any other section. Please ensure that your ethics statement is included in your manuscript, as the ethics statement entered into the online submission form will not be published alongside your manuscript.

8. We note you have included a table to which you do not refer in the text of your manuscript. Please ensure that you refer to Table 5 in your text; if accepted, production will need this reference to link the reader to the Table.

Reviewers' comments:

Reviewer's Responses to Questions

**Comments to the Author**

1. Is the manuscript technically sound, and do the data support the conclusions?

Reviewer #1: No

Reviewer #2: Yes

Reviewer #3: Yes

2. Has the statistical analysis been performed appropriately and rigorously? 

Reviewer #1: No

Reviewer #2: Yes

Reviewer #3: Yes

3. Have the authors made all data underlying the findings in their manuscript fully available?

Reviewer #1: No

Reviewer #2: Yes

Reviewer #3: Yes

4. Is the manuscript presented in an intelligible fashion and written in standard English?

Reviewer #1: Yes

Reviewer #2: Yes

Reviewer #3: Yes

5. Review Comments to the Author

Reviewer #1: Dear auturs:

The manuscript addresses a topic of interest; however, it does not yet meet the standards required for publication.

Technical soundness: The study lacks sufficient methodological rigor. Key details regarding experimental design, controls, replication, and sample size justification are insufficiently described, which limits confidence in the validity of the findings.

Statistical analysis: The statistical analyses are not rigorous enough to support the conclusions. Analyses are largely descriptive, with limited justification of statistical tests and absence of effect sizes or confidence intervals.

Data–conclusion alignment: Several conclusions appear overstated and are not fully supported by the presented data. Conclusions should be more cautious and strictly aligned with the results.

Presentation and clarity: Improvements are needed in data presentation (figures/tables) and overall clarity to allow readers to better interpret the findings.

Substantial revision, including strengthening the methodology, improving statistical rigor, and revising conclusions, would be required before the manuscript could be reconsidered.

Reviewer #2: The current may consider for publication. Check the references list and text reference should be matched and proper format of the journal.. The above manuscript have good information about the tick control through natural products.

Reviewer #3: I thank the authors for their efforts in preparing this manuscript, which addresses a relevant topic in parasite control using plant-based, environmentally friendly alternatives, but the study in its current form suffers from methodological, statistical, and editorial weaknesses that require substantial revision before acceptance for publication.

6. PLOS authors have the option to publish the peer review history of their article (what does this mean?). If published, this will include your full peer review and any attached files.

Reviewer #1: No

Reviewer #2: **Yes:** Dr Sachin Kumar (Ph.D, NPDF)

Reviewer #3: **Yes:** Mohamed Mahmoud Baz

---

## [Author Response · Author response to Decision Letter 1]

18 Mar 2026

Editor comments

AU: Fixed

AU: The information regarding the ethical permit and ethical statement has now been incorporated into the Methods section.

3. Please provide additional details regarding consent obtained from the owners of the animals and mention the detailed procedure.

AU: Participant consent has been included in the Methods section within the ethical statement.

4. Please note that funding information should not appear in any section or other areas of your manuscript. We will only publish funding information present in the Funding Statement section of the online submission form. Please remove any funding-related text from the manuscript.

AU: Fixed

“The authors would like to express their sincere gratitude to the animal owners for their cooperation. We also wish to acknowledge the community animal health workers in the study area for their invaluable assistance in facilitating the fieldwork. Finally, we extend our thanks to the Amhara Region Agricultural Research Institute (ARARI) for the financial and logistical support.”

AU: The recommendation has been noted, and the statement has been corrected/inserted.

AU: The statement has now been included in the cover letter.

6. In the online submission form you indicate that your data is not available for proprietary reasons and have provided a contact point for accessing this data. Please note that your current contact point is a co-author on this manuscript. According to our Data Policy, the contact point must not be an author on the manuscript and must be an institutional contact, ideally not an individual. Please revise your data statement to a non-author institutional point of contact, such as a data access or ethics committee, and send this to us via return email. Please also include contact information for the third party organization, and please include the full citation of where the data can be found.

AU: The Data Availability Statement has been revised, and the raw data used in the analysis have now been uploaded to the system.

7. Your ethics statement should only appear in the Methods section of your manuscript. If your ethics statement is written in any section besides the Methods, please move it to the Methods section and delete it from any other section. Please ensure that your ethics statement is included in your manuscript, as the ethics statement entered into the online submission form will not be published alongside your manuscript.

AU: The ethical statement has now been moved to the Methods section.

8. We note you have included a table to which you do not refer in the text of your manuscript. Please ensure that you refer to Table 5 in your text; if accepted, production will need this reference to link the reader to the Table.

AU: Fixed

AU: No reviewer recommended this claim.

Reviewer #1 comments

The manuscript addresses a topic of interest; however, it does not yet meet the standards required for publication. Technical soundness: The study lacks sufficient methodological rigor. Key details regarding experimental design, controls, replication, and sample size justification are insufficiently described, which limits confidence in the validity of the findings.

AU: The methodology section has been significantly improved based on this comment, and the manuscript is now written in detail.

Statistical analysis: The statistical analyses are not rigorous enough to support the conclusions. Analyses are largely descriptive, with limited justification of statistical tests and absence of effect sizes or confidence intervals.

AU: The statistical analysis of some results has been revised, and the suggestion has been incorporated.

Data–conclusion alignment: Several conclusions appear overstated and are not fully supported by the presented data. Conclusions should be more cautious and strictly aligned with the results.

AU: We have revised the conclusion to ensure it is solely data-driven and precise.

Presentation and clarity: Improvements are needed in data presentation (figures/tables) and overall clarity to allow readers to better interpret the findings. Substantial revision, including strengthening the methodology, improving statistical rigor, and revising conclusions, would be required before the manuscript could be reconsidered.

AU: Revisions have been made in the indicated sections in accordance with the general comments provided.

Reviewer #2: The current may consider for publication. Check the references list and text reference should be matched and proper format of the journal. The above manuscript have good information about the tick control through natural products.

AU: Thank you for endorsing the manuscript for publication.

Reviewer #3: I thank the authors for their efforts in preparing this manuscript, which addresses a relevant topic in parasite control using plant-based, environmentally friendly alternatives, but the study in its current form suffers from methodological, statistical, and editorial weaknesses that require substantial revision before acceptance for publication.

Comments and Suggestions for Authors

Title

- The title is direct and good but delete the word “Tick Species” is unnecessary because the

species is already defined.

AU: Fixed

Abstract

- The LC50 or LC90 value not mentioned, which is essential in toxicity studies

AU: Thank you for your valuable insights. The LD₅₀ and LD₉₀ have now been analyzed and incorporated into the Results section, and this comment has substantially improved the manuscript.

- The phrase “These findings support the reason for the traditional use; Please delete it

AU: Deleted

- There is no mention of the tick exposure period (7 days) for understanding the reported

mortality rates.

AU: Ticks were exposed for 10 minutes in a 10ml container. The duration of the experiment was seven days, with observations at three-hour intervals.

Introduction

What new compared to previous studies?

AU: Some previous reports on acaricide resistance experiments have been incorporated, and the driving force for this study was the widespread use of plant materials for tick control alongside the emergence of resistance to commercial acaricides.

Is this the first study in Waghimra?

AU: Yes, of course. The problem statement was that, despite the common practice of using plants for tick control in the area, there was no experimentally tested evidence supporting their efficacy.

Are A. variegatum ticks considered a new record in Waghimra?

AU: This tick species is the most abundant and frequently reported.

Materials and methods

- Why was the study described as cross-sectional; when it was experimental, not observational? Please review.

AU: The experiment has been clarified as a seven-day laboratory trial, and the field data collection as a cross-sectional survey conducted in May 2023. The statement has now been rephrased and corrected.

- Why was the study long-term (May 2023–September 2023)? Was it seasonal?

AU: the year is typological error and corrected as May 2023 to June 2023.

- The plants used in the study were not classified.

AU: The plant family classifications have been added to Table 1.

- Further clarification is needed in the Crud extraction method and storage section.

AU: The procedures for crude extraction and storage have now been clarified.

- Storing the extract for 5 months is excessive. Why was this statement included?

AU: Thank you for pointing out the inappropriate description of sample storage. The statement has now been revised, as the original context was found to be incorrect—a mistake that occurred during contextualization.

- Was there a positive control in the juvenile stage experiment?

AU: There was no a positive control, instead the negative control was used (distilled water).

- Statistical analysis: No probit regression was used to calculate the LC50.

AU: Thank you for the insightful comment. The manuscript has greatly benefited from it, and the probit analysis has now been incorporated and LD50 and LD90 is included.

Results & Dissection.

- The highest mortality rate (70%) is considered insignificant (low).

AU: The claim is correct; however, this efficacy is even higher than that of the commercial acaricides diazinon and amitraz, which were reported in our previous studies at 58% and 38%, respectively. https://doi.org/10.1016/j.vprsr.2023.100885

- There is a discrepancy in the results, where in the ANOVA table, it states “Dose” = Not

Significant (0.083; yet the text mentions significant dose-dependent killing effect; Please

AU: Fixed.

- The results were compared with studies on Rhipicephalus microplus and Tetranychus urticae.

This is a scientific error because different species have different sensitivities. The discussion

should first focus on the same species and then, in parallel, cite other species.

AU: First, we appreciate the insightful comment that different tick species could exhibit varying levels of sensitivity. However, our systematic search revealed that most reported results were not species-specific or primarily focused on Rhipicephalus microplus, as this species is the most important in many regions, including South America and Africa.

, - 98% inhibition (percent control (PC)) was recorded, but there is no calculation of the Reproductive Index or Egg Hatchability Assessment.

AU: The “Reproductive Index or Egg Hatchability Assessment” assessment was not conducted, as the experiment was specifically designed to evaluate direct tick mortality and oviposition inhibition effects. The percentage control (PC) was calculated, as described in the Methods section, to determine the reduction in egg laying. Additionally, this comment, along with the absence of a positive control, has been included as a limitation of the study.

---

## [Decision Letter · Decision Letter 1]

9 Apr 2026

Acaricidal Activity of Plant Extracts against Amblyomma variegatum in Waghimra, Northeastern Ethiopia

PONE-D-25-64821R1

Dear Dr. tibebu,

We’re pleased to inform you that your manuscript has been judged scientifically suitable for publication and will be formally accepted for publication once it meets all outstanding technical requirements.

Kind regards,

Kandasamy Ulaganathan

Academic Editor

PLOS One

Additional Editor Comments (optional):

Reviewers' comments:

Reviewer's Responses to Questions

**Comments to the Author**

1. If the authors have adequately addressed your comments raised in a previous round of review and you feel that this manuscript is now acceptable for publication, you may indicate that here to bypass the “Comments to the Author” section, enter your conflict of interest statement in the “Confidential to Editor” section, and submit your "Accept" recommendation.

Reviewer #2: All comments have been addressed

Reviewer #3: All comments have been addressed

2. Is the manuscript technically sound, and do the data support the conclusions?

Reviewer #2: Yes

Reviewer #3: Yes

3. Has the statistical analysis been performed appropriately and rigorously? 

Reviewer #2: Yes

Reviewer #3: Yes

4. Have the authors made all data underlying the findings in their manuscript fully available?

Reviewer #2: Yes

Reviewer #3: (No Response)

5. Is the manuscript presented in an intelligible fashion and written in standard English?

Reviewer #2: Yes

Reviewer #3: Yes

6. Review Comments to the Author

Reviewer #2: After revision the current manuscript is considerably for publication.. The research paper is good for all the researchers who working on tick research.

Thank you

Reviewer #3: Dear Authors,

I commend the authors for their excellent work and appreciate their effort addressing the reviewers' comments, resulting in a manuscript now ready for publication. I would only ask the authors to conduct a final review of the manuscript before it goes to print to correct some linguistic errors. This does not negate the manuscript's readiness and acceptance for publication.

- Please abbreviate the names of plant and animal species after mentioning them for the first time.

- After accepting the track change corrections, please adjust the spacing and note that some text is written in bold.

The manuscript is adequate and can be accepted as is, and it doesn't need it back

7. PLOS authors have the option to publish the peer review history of their article (what does this mean?). If published, this will include your full peer review and any attached files.

Reviewer #2: **Yes:** Dr Sachin Kumar

Reviewer #3: **Yes:** Mohamed Mahmoud Baz

---

## [Editor Report · Acceptance letter]

PONE-D-25-64821R1

PLOS One

Dear Dr. tibebu,

I'm pleased to inform you that your manuscript has been deemed suitable for publication in PLOS One. Congratulations! Your manuscript is now being handed over to our production team.

Kind regards,

on behalf of

Dr. Kandasamy Ulaganathan

Academic Editor

PLOS One